# Recovery of Mycobacteria from Heavily Contaminated Environmental Matrices

**DOI:** 10.3390/microorganisms9102178

**Published:** 2021-10-19

**Authors:** Vit Ulmann, Helena Modrá, Vladimir Babak, Ross Tim Weston, Ivo Pavlik

**Affiliations:** 1Public Health Institute in Ostrava, Partyzanske Nam. 7, 702 00 Ostrava, Czech Republic; vit.ulmann@zuova.cz; 2Faculty of Regional Development and International Studies, Mendel University in Brno, Tr. Generala Piky 7, 613 00 Brno, Czech Republic; helena.modra@mendelu.cz; 3Veterinary Research Institute, v.v.i., Hudcova 70, 621 00 Brno, Czech Republic; babak@vri.cz; 4Department of Biochemistry and Genetics, La Trobe Institute for Molecular Science, La Trobe University, Bundoora, Melbourne, VIC 3086, Australia; R.Weston@latrobe.edu.au

**Keywords:** non-tuberculous mycobacteria (NTM), mycobacteria other than tuberculosis (MOTT), saprophytic environmental mycobacteria, decontamination, quaternary ammonium compounds

## Abstract

For epidemiology studies, a decontamination method using a solution containing 4.0% NaOH and 0.5% tetradecyltrimethylammonium bromide (TDAB) represents a relatively simple and universal procedure for processing heavily microbially contaminated matrices together with increase of mycobacteria yield and elimination of gross contamination. A contamination rate only averaging 7.3% (2.4% in Cluster S; 6.9% in Cluster R and 12.6% in Cluster E) was found in 787 examined environmental samples. Mycobacteria were cultured from 28.5% of 274 soil and water sediments samples (Cluster S), 60.2% of 251 samples of raw and processed peat and other horticultural substrates (Cluster R), and 29.4% of 262 faecal samples along with other samples of animal origin (Cluster E). A total of 38 species of slow and rapidly growing mycobacteria were isolated. *M. avium* ssp. *hominissuis*, *M. fortuitum* and *M. malmoense* were the species most often isolated. The parameters for the quantitative detection of mycobacteria by PCR can be significantly refined by treating the sample suspension before DNA isolation with PMA (propidium monoazide) solution. This effectively eliminates DNA residue from both dead mycobacterial cells and potentially interfering DNA segments present from other microbial flora. In terms of human exposure risk assessment, the potential exposure to live non-tuberculous mycobacteria can be more accurately determined.

## 1. Introduction

### 1.1. Genus Mycobacterium

Mycobacteria are acid-fast, Gram-positive bacilli characterised as intracellular parasites. Currently, 195 mycobacterial species and subspecies have been identified (List of Prokaryotic Names with Standing in Nomenclature, LPSN) [1]. Similar to other groups of bacteria, the number of described mycobacterial species has undergone an expansion due to the high discriminatory nature of whole genome sequencing [2]. The group includes obligate pathogenic mycobacteria causing tuberculosis (TB; members of the *Mycobacterium tuberculosis* complex, MTC) and leprosy (*M. leprae*), as well as non-tuberculous mycobacteria (NTM) occurring widely in aquatic and terrestrial environments [3,4]. As classical TB still poses a significant worldwide threat to human health, infections caused by NTM are increasing nowadays. The trend for the increase in the incidence of NTM infections is connected with aging populations, immunosuppressive therapy, the emergence of AIDS, and an increase in surveillance [3]. Infections represent a serious threat to human health and result in high costs associated with the adequate care of immunocompromised patients.

Within the NTM, the *M. avium* complex (MAC) is of most significance and it also belongs to the most studied NTM. It is known that MAC strains bearing plasmids associated with virulence can be aerosolised, suggesting a possible mechanism for airborne acquisition of these organisms [5]. The rapidly growing *M. fortuitum* is an opportunistic NTM that has been identified in several nosocomial infections involving hospital instruments, hospital water systems, and peritoneal catheters [6]. Currently, multidrug-resistant strains of *M. abscessus* pose a major threat, especially in patients with cystic fibrosis, when adequate therapy is no longer available [7]. During the last decade, epidemiological situation concerning mycobacterioses deteriorated, especially due to changes in lifestyle. The urbanisation process was followed by suburbanisation, and eating habits have changed (raw food, not properly cooked food products and/or bio products are preferred), population aging process has been observed, cystic fibrosis and chronic obstructive pulmonary disease (COPD) are leading to the increase in serious comorbidities, and mass BCG vaccination in children has been discontinued in developed countries. All these risk factors are connected with increase of NTM infections in humans [8,9]. From extrapulmonary infections, lymphadenitis in children predominated over the last three years [10]. The slow-growing *M. marinum* is the causal agent of chronic systemic infections in fish and it occasionally causes granulomatous skin lesions involving the hands, forearms, elbows, and knees in humans reporting contact with aqueous environments [11,12].

### 1.2. Direct Recovery of Mycobacteria—Methods in Microbiology

Direct detection of mycobacteria consists in microscopy, cultivation, and capture of genetic material. The capture of viable mycobacteria is hampered by their very slow growth. Due to the slow metabolism, energy-intensive construction of the cell wall and inefficient genetic transfer systems lead to a generation time of more than 24 h. Visible colony growth can be expected in most species, usually after more than a week. Therefore, any non-sterile material must be freed of other faster-growing microflora before cultivation. In this case, the relative resistance of mycobacteria to chemical agents is used—most often lye, as well as organic and inorganic acids or detergents. Historically, decontamination procedures have been developed mainly for the field of human diagnostics. For decontamination of clinical material samples from the respiratory tract, a method combining *N*-acetylcysteine and 2% sodium hydroxide or a 4% NaOH solution are used as standard [13,14].

Microbially highly loaded specimens, such as the stool or sputum of cystic fibrosis patients, are decontaminated with oxalic acid [15]. Urine can be primarily decontaminated with 2% hydrochloric acid. It is also possible to use the swab method [16]. For cultivation, the most used media are classical Löwenstein Jensen and Middlebrook solid media or liquid Middlebrook broth with or without enrichment with antibiotics and growth supplements. Enriched or special media for automated cultivation are already more economically demanding for processing a larger number of samples and can in some cases limit the growth of some environmental, mainly psychrophilic or thermophilic species [17,18,19]. The list of works dealing with the direct detection of mycobacteria in both natural and human-influenced environments is more modest.

Most attention is paid to historically proven findings and in terms of exposure to significant water sources. Domestic tap water, but especially water for the hygiene of employees in industry or patients of clinical facilities, requires special attention and constant quality monitoring. Resistance to surfactants, mainly quaternary ammonium salts (QUAS), is used for direct detection of mycobacteria in water. The standard technique is to use a combination of 1% lauryl sulphate and 2% lye for less microbially loaded samples (water less chemically and physically treated) or 0.005% cetylpyrimidium chloride (CPC) solution [20]. There is very little work on the development of methods for the decontamination of solid, very microbially loaded matrices. In similar studies aimed at detecting *M. ulcerans* in tap water samples in Ghana, this method was used successfully [21]. A two-step decontamination process using a sodium hydroxide solution and cetrimide was used to detect mycobacteria from soil samples in India [22].

Resistance of mycobacteria to relatively high concentrations of QUAS has long been monitored in relation to nosocomial infections [23] and the risk of developing mycobacterial strains resistant to surface sterilisation. QUASs have minimal mycobactericidal activity in short-term exposures. Used alone or in combination with other antiseptics, QUASs’ activity is generally limited to other microbes, particularly in the presence of high organic pollution [24]. Therefore, a combination of chemical agents including a QUAS should increase effectiveness of a decontamination process prior to mycobacteria culture examination.

### 1.3. Biomolecular Methods

The quantitative polymerase chain reaction (qPCR) method was chosen for a quick preliminary assessment of the presence of mycobacteria. This relatively easily accessible method is currently being used for basic mycobacteria monitoring in various matrices. qPCR can be used for accurate detection of mycobacteria with partial quantification, as verified by several studies. These studies focused on the detection of NTM in multiple water samples [20,25,26,27,28,29,30], as well as the detection of obligate and potentially pathogenic mycobacteria in clinical samples from humans [31], tissue samples from animals [32,33,34,35,36], and in food and feed samples [33,37,38,39,40,41].

Viable plate counts represent the most sensitive method for detection of viable bacteria [25] and are still regarded as the gold standard method. However, their application in indolent microorganisms (including mycobacteria) is difficult, time-consuming, and may be over burdened with false negative results, especially in mycobacteria culturing due to the presence of antibiotics used to suppress relatively rapidly growing contaminating microflora [42].

The recent development of using viability dyes in conjunction with quantitative PCR (viability qPCR or v-qPCR) has been demonstrated to be effective in identifying and quantifying viable microorganisms in complex samples [43,44] and could potentially be an alternative approach to classical plate counts. For these assays, viability is based on membrane integrity and the ability of chemical compounds to bind to DNA (v-qPCR dyes). In cells or virions with a compromised envelope or membrane due to the application of heat, freeze, or chemical compounds, a v-qPCR dye can penetrate and covalently bind to nucleic acids in the presence of light and consequently prevent the amplification of that DNA in subsequent real-time quantitative PCR. In viable cells, with an intact envelope, the v-qPCR dye cannot penetrate inside and thus its DNA is free for amplification in a subsequent qPCR assay. A number of v-qPCR dyes have been developed for viability studies: EMA (ethidium monoazide), PMA (propidium monoazide) [45], and TOMA (thiazole orange monoazide) [46]. The utility of the PMA v-qPCR method has been demonstrated for evaluating the efficiency of water disinfection, sensitivity of *M. avium* ssp. *paratuberculosis* to certain chemical treatments, and quantification of *M. tuberculosis* and other clinically relevant mycobacterial species in human and animal samples [47,48,49,50].

PMA (phenanthridium, 3-amino-8-azide-5-[3-(diethymethylammonio) propyl]-6-phenyl dichloride) is the most commonly used and most examined v-qPCR dye and has recently been developed into a commercial product that is used in mycobacteriological laboratories [51,52,53]. This method for mycobacteria detection was developed and optimised specifically for drinking and surface water and water biofilms [20,25,26,27,28,29]. An effective and rapid method for the detection of viable mycobacteria from heavily contaminated environmental matrices has not yet been described.

Thus, the overall aim of this study was to improve the process of mycobacteria detection by culture and qPCR in heavily contaminated environmental samples. Much of the ecology of mycobacterial species remains undetected, as well as their presence in the natural environment and the routes of transmission to the immediate surroundings of man. Exposure risks and some epidemiological contexts are gradually being revealed in our ongoing study.

## 2. Materials and Methods

During the course of an extensive ecological–epidemiological study, samples of peat, peat substrates, soil, animal faeces, wastewater, and watercourse sediments were collected for assessment as potential sources of pathogenic and NTM. Large quantities of samples were collected per day, and consequently an optimised method to process samples and test for the presence of mycobacteria was required to accommodate such a large intake at once. Although extensively used for clinical samples, decontamination with a sodium hydroxide solution requires a long incubation step that was not suitable for these types of samples, where the processing for this study was required to be completed within 24 h of collection. In the relevant literature there was no consensus as to the most efficient way to process environmental samples for mycobacterial culture in the required time constraints [25,26].

### 2.1. Samples’ Collection

A total of 787 samples of different environmental matrices were collected in the territory of Moravia as a part of the Czech Republic in period from January 2019 to March 2021. These samples were collected as part of a study to monitor the spread of mycobacteria and evaluate the species spectrum of mycobacteria in the natural environment, especially in karst and protected landscape locations, as well as area subject to mining activities. These samples were divided in to three main clusters:

Cluster S (274 soil and water sediment samples of rivers and water bodies and sewage sludge from wastewater treatment plants collected in man-made systems, settlements and municipalities, industrial and mining areas, artificial bathing tanks, and related and adjacent rivers and streams).

Cluster R (251 samples of raw and processed peat originating from forest peatlands, gardens, and commercial horticultural products).

Cluster E (262 faeces and products from animals including birds, bats, and other mammals).

### 2.2. Pre-Treatment Procedure

Material weighing from 5 to 10 g (with a maximum volume of 15 mL) was resuspended in 20 mL of distilled water containing 1% Tween 80 and 20 glass beads (2 mm diameter). Suspension was homogenised thoroughly by vortexing for 1 min, followed by shaking at 300 oscillations/min for 15 min. The homogenised samples were centrifuged at 500 revolutions per minute (rpm) for 10 min. The supernatant was collected and centrifuged at 4300 rpm; this supernatant was then discarded, and the pellet was subsequently used.

### 2.3. Decontamination with NaOH and TDAB

Fresh decontamination solution was prepared containing 4% NaOH (Lach-Ner, Neratovice, Czech Republic) and 0.5% TDAB (tetradecyl-trimethylammonium bromide; Duchefa Biochemie B.V, Haarlem, The Netherlands), stable for 14 days at 4 °C. The pre-treated pellet was resuspended in 10 mL of decontamination solution, vortexed, and then homogenised on a shaker for 15 min. After centrifugation for 20 min at 4300 rpm, the supernatant was discarded, and the pellet resuspended in 15 mL of distilled water; importantly, this decontamination step did not exceed 40 min. The homogenisation and centrifugation steps were repeated, and the pellet was resuspended in 2.5 mL of phosphate buffer (pH 6.8) and thoroughly mixed and immediately submitted for culturing. Four Löwenstein–Jensen media slants were inoculated with 0.2 mL (0.8 mL total) of processed sample and placed in pairs in a thermostat at 30 and 37 °C and left to incubate for 12 weeks, with growth assessments performed weekly. A 0.8 mL aliquot of processed sample was taken in a separate 1.5 mL Eppendorf tube for qPCR.

### 2.4. Isolates Identification

Suspected mycobacterial isolates were examined macroscopically and subsequently by microscopy with Ziehl–Neelsen staining. Preliminary identification was made by GenoType^®^ Mycobacterium CM/AS (Hain Lifescience GmbH, Nehren, Germany). Slightly contaminated and suspected mixed isolates were re-cultured, and secondary isolates were subjected to further identification. The DNA of isolates was extracted and used as template for PCR amplification of the 16srRNA and *hsp65* genes using the universal bacterial primers 5′CCT ACG GGN GGC WGC AG3′ and 5′GAC TAC HVG GGT ATC TAA TCC3′ of the V3 and V4 variable regions [54] and *Mycobacterium hsp65* primers 5’ACC AAC GAT GGT GTG TCC AT3’ and 5’CTT GTC GAA CCG CAT ACC CT3’ [55], respectively. The resulting amplicons obtained by both of the methods were purified and sequenced by Eurofins (Ebersberg, Germany). Identification of mycobacterial species was performed by BLAST analysis. Isolates belonging to *M. avium* complex members were further identified by the PCR method for the detection of the IS*901* amplicon specific for *M. avium* ssp. *avium* and the IS1245 amplicon specific for *M. avium* ssp. *hominissuis* [37].

Evaluation of accompanying microflora (contamination): All cultures were evaluated microscopically, and massive growth after one day of cultivation was inoculated with a loop by cross-smearing on blood agar and further cultured. From the solid media, which were degraded by the proteolytic activity of the bacteria to a fluid, 100 μL was carefully removed with a Pasteur pipette and inoculated with a line and smeared on blood and Endo agars (for the selection of G- microbes). The media were cultured at 30 and 37 °C. Growth was assessed after 24 h and the culture was closed after 5 days. Individual colonies were identified by MALDI-TOF (MALDI Biotyper, Bruker-Daltonics Billerica, MA, USA).

### 2.5. qPCR Method

This qPCR method was selected as a reference method for the evaluation of mycobacterial presence using the commercial Real-time PCR Z-Path-Mycobacterium_spp detection kit for *Mycobacterium* (Primerdesign Ltd., Camberley, UK). DNA was isolated from 0.8 mL of decontaminated sample by centrifugation at 14,000 rpm/min, discarding the supernatant and then processing the pellet with the DNA E.Z.N.A.^®^ Soil DNA Kit (Omega Bio-tek, Norcross, GA, USA) according to manufacturer’s instructions. A total of 0.5 µL of DNA isolation product was added to a Precision PLUS 2× qPCR Master Mix (Primerdesign Ltd., Camberley, UK), and the qPCR reaction was run with a CFX96 real-time PCR detection system (Bio-Rad Laboratories, Hercules, CA, USA) using the following thermocycler conditions: enzyme activation at 95 °C for 2 min, 50 cycles at 95 °C for 10 s for denaturation and at 60 °C for 60 s for aneling and data reading.

### 2.6. Viability Testing by PMA Treated DNA

An alternative approach using a PMA qPCR method was tested to determine the viability of mycobacteria after the decontamination process. A total of 100 μL of raw and decontaminated suspensions (after neutralisation) were treated with suspension of 50 μM PMA (Biotum Inc. Hayward, CA, USA) diluted in 20% dimethyl sulfoxide (DMSO; Sigma-Aldrich, Burlington, MA, USA) and incubated in the dark for 5 min; then, the mixture was exposed to light from a 650 W halogen bulb for 2 min [48]. Cells were lysed, DNA was isolated, and qPCR was performed as described previously (Section 2.5) to determine the viability of cells. Viability determination is based on two separate qPCR reactions.

### 2.7. Statistical Analysis

Calculation of the average number of colony-forming units (CFU) per 5 g of solid sample: The average CFU per media was first calculated by dividing the CFU count by the number of medium with visible growth. If the strain did not grow at a certain temperature, the results were calculated only from the data of those incubation temperatures at which the strain was able to grow. CFU per gram of solid sample was then calculated using the appropriate dilution factors. For statistical processing, soil colony counts were generalised to the following ranges: 1–10 CFU, 11–100 CFU, and >100 CFU (too numerous to count = TNTC) for continuous colonies and confluent growth (Figure 1).

Curves for quantitative PCR were derived from the appropriately diluted internal standard included in a qPCR kit. Results are represented as relative fluorescent units (RFU), approximated according to a measured standard by CFX-manager 2.0. software (Bio-Rad Laboratories, Hercules, CA, USA). Data analysis was performed using statistical software Statistica 13.2 (StatSoft Inc., Tulsa, OK, USA). *p*-values less than 0.05 were considered statistically significant. For multiple comparison tests, Bonferroni adjustment of *p*-values were used. Simple summary calculations were performed in the MS Excel program (Microsoft, Redmont, WA, USA).

## 3. Results

This section is divided by subheadings in two parts: the first part dealing with culture results and the second part is devoted to the qPCR results.

### 3.1. Culture Results

#### 3.1.1. Culture Contamination Rates

Contamination of a culture occurred when a complete deterioration of all culture media of one sample by massive coverage of the media surface with a non-mycobacterial microbe or its complete degradation (liquefaction) of the media. The overall rate of contamination by the total volume of processed samples was 7.4% (*n* = 787). Differences in the degree of contamination were recorded within and between three individual functional clusters. The lowest media contamination of 2.4% (*n* = 251) was detected in Cluster R, representing raw plant material and peat, followed by 6.9% (*n* = 274) in contaminated water sediment samples, including sewage sludge from wastewater treatment plants with contamination. The most degraded cultures were recorded in the third Cluster E containing animals’ faeces with 12.6% (*n* = 262) contaminated samples (Figure 1).

The contaminating microflora was thoroughly examined by microscopy and growth morphology on blood agar. The most abundant contaminating species identified: from a heavily decomposed (liquefied and previously solid) media with odorous, pressured gaseous content, we microscopically observed a massive abundance of *Clostridium* spp. rods; the other most frequent species revealed were *Bacillus* spp., *Proteus* spp. and *Paenibacillus* spp. Contaminants from samples from Cluster R consisted of *Bacillus* spp., *Pseudomonas* spp., *Actinomyces* spp., *Staphylococcus* spp., *Clostridium* spp. and *Serratia* spp. Cluster S samples had contamination from *Pseudomonas* spp., *Bacillus* spp., *Paenibacillus* spp., *Micrococcus* spp., *Corynebacterium* spp. and *Flavobacterium* spp. In Cluster E, with the most occurrences of contaminated samples (12.6%) were *Clostridium* spp. rods, *Bacillus* spp., *Proteus* spp., *Paenibacillus* spp., *Corynebacterium* spp. and *Flavobacterium* spp. and other coliform Gram-negative rods were also detected (Figure 1).

#### 3.1.2. Yield of Mycobacteria by Cultivation Method

Detection of mycobacteria was successful in 39% of samples (*n* = 787) by culture (Figure 2). The highest detection rate of mycobacteria was 60.2% (*n* = 251) samples from Cluster R (peat and plant material), followed by Cluster S (water sediments etc.) with 28.5% (*n* = 274) and the least in Cluster E (animals’ faeces etc.) with 29.4% (*n* = 262) (Figure 2).

A total of 506 isolates belonging to 38 mycobacterial species (102 mycobacterial isolates could not be identified to the species or complex level) were isolated, and the results are summarised in Table 1. The three most common species identified were *M. avium* ssp. *hominissuis* (155 isolates), *M. fortuitum* and *M. malmoense* (39 isolates each). In terms of mycobacterial species diversity, the widest mycobacterial species spectrum (25 different mycobacterial species) was found in Cluster S (water sediments etc.). In this cluster, the three the most often detected species were *M. avium* spp. *hominissuis* (20 isolates), *M. fortuitum* (13 isolates), and *M. paragordonae* (8 isolates). Cluster R (peat and plant material) had the next highest species diversity with 18 mycobacterial species identified. The three the most often detected species from this cluster were *M. avium* ssp. *hominissuis* (103 isolates), *M. malmoense* (38 isolates), and *M. xenopi* (37 isolates). The lowest species diversity was observed in Cluster E (faeces etc.) with 14 mycobacterial species, with the most detected species *M. avium* spp. *hominissuis* (20 isolates), *M. fortuitum* (17 isolates)*, M. hiberniae*, *M. peregrinum* and *M. terrae* complex (5 isolates each).

#### 3.1.3. Colony-Forming Unit (CFU) Counts Detected by Culture Method

The CFUs per 5.0 g of examined matrix were calculated as described in the Materials and Methods section after 12 weeks of culture. For statistical comparison and evaluation of matrices, the geometric mean of the number of CFUs of all positive cultures in individual clusters was determined (Table 2).

The highest presence of viable mycobacterial cells was recorded in samples from Cluster R (peat and plant material). The geometric mean CFU count of this cluster was 69.20, with the 10 samples with the greatest number of viable cells recorded being from this cluster. Tens to hundreds of CFUs were detected in most of the positive samples from Cluster R. In clusters E and S, the average number of CFUs were 10.48 and 14.21, respectively (Figure 3).

### 3.2. qPCR Results

#### 3.2.1. Mycobacteria Quantification in Different Matrices by qPCR

Within the individual clusters, the RFU values of positive samples were averaged for statistical evaluation, and the geometric mean of RFU of the individual clusters was determined. Mycobacterial DNA was detected in over 73% of samples from the Clusters R and E and 57% of samples from Cluster S. Compared to Clusters S and E, positive samples from Cluster R had a sixfold greater RFU median value and approximately fourfold average RFU value (Table 3). Average RFU for each cluster is visualised in Figure 4.

#### 3.2.2. Parameters of the qPCR Method

A basic comparison of culture and qPCR results was performed by calculating a sensitivity of the qPCR method of 92.2% and a specificity of 49.9% (Table 4 and Figure 5).

#### 3.2.3. Model of Predictive Values of qPCR

For the construction of model for the prediction of culture results based on the mycobacteria absolute numbers gained by qPCR method, we performed binary logistic regression analysis. For the modelling of a probability of mycobacteria detection by culture based on the log-transformed values of qPCR method, all samples were taken into consideration. Culture results were considered as a dichotomised variable that can take the value of “0” (negative culture) or “1” (culture - positive), irrespective of the cultured CFU numbers. The binary logistic regression in a common form with two-parametric logit function was used.

Suitability of the model was confirmed by the Hosmer–Lemeshow goodness-of-fit test (*p* > 0.05) and its statistical significance proven for both the whole model (likelihood ratio test; *p* < 0.01) and parameters of a model (Wald statistic; *p* < 0.01). According to the model, it is possible to estimate the probability of positive culture using absolute number of MYC as derived from qPCR. For example, in samples with 105 mycobacterial cells, there is approximately 74.0% probability that these will be positive by culture (Figure 5).

The odds ratio value derived from the model used in this study was 2.08 (95% CI: 1.88 to 2.31). This shows that odds of the culture positivity increased more than two times when qPCR value increased by one order of magnitude. The logarithmic cut-off value derived from the model was determined to be 3.2428, i.e., 1.75 × 10^3^ mycobacterial cells, as measured by the qPCR method (Figure 6). After application of this cut-off value on the raw data, we were able to correctly identify 78.8% of culture-positive (diagnostic sensitivity) and 77.3% of culture-negative samples for mycobacteria (diagnostic specificity; Table 5).

#### 3.2.4. Evaluation of the Used PMA-qPCR Method

Parallel quantification of mycobacterial DNA by PMA-qPCR was performed on 25 faecal samples. A significant order of magnitude decrease in RFU values (Figure 7) and a higher degree of correlation in negativity compared to culture were observed. The sensitivity and specificity of PMA-qPCR reached 100% and 80%, respectively, when compared to culture examination (Table 6).

## 4. Discussion

The primary goal of this work was to determine an optimal and importantly, universal decontamination procedure in order to enable the processing of a wide range of different matrices. Standard decontamination methods for the processing samples of clinical material with NaOH and *N*-acetyl cysteine could not be used for highly contaminated samples due to the presence of certain environmental micro-organisms (i.e., sporogenic bacteria and moulds). According to previously published data [56,57], a combination of different chemical substances would be required.

A common decontamination method using HCl and oxalic acid was also not tested due to previously published worked that indicated it results in low yield of mycobacteria and high proportion of media contamination [58]. Quaternary ammonium salts were considered and investigated for decontamination, but the following two technical complications were encountered. When decontaminating faecal samples, it was difficult to obtain (1) a homogeneous sample for inoculation and (2) an aliquot for the use of molecular biological method (qPCR). After decontamination with quaternary ammonium salts and final centrifugation, sparingly soluble precipitates often formed, and a large amount of foaming of the processed sample often occurred (unpublished observation), causing the aforementioned issues.

Initially for this study, CPC surfactant was used with 4% NaOH for the decontamination of samples. In these experiments, the decontamination efficiency was not satisfactory because the contamination rate of the culture media exceeded 30% (unpublished data). A “pre-cultivation” method published by Portaels et al. [59] was also tested. In this method, contaminating microbes are revived and then subsequently eliminated. However, this procedure was unsuccessful, with up to 80% contamination and massive multiplication of other microbes, which made it impossible to isolate mycobacteria (unpublished data). Allen [60] also noted similarly high sample contamination when decontaminating human faecal samples from patients with human tuberculosis.

Therefore, we developed a new decontamination procedure using a mixture of 4.0% NaOH and 0.5% TDAB. Löwenstein–Jensen media was used for culturing due to its widespread use for culturing mycobacteria. A 4.0% NaOH solution is an established decontamination solution for isolating mycobacteria, and the addition of TDAB enhanced isolation of mycobacteria; however, it was not possible to rule out any possible detrimental (even devitalising) effects that TDAB may have on living mycobacteria present in the examined sample.

Due to the nature and source of the examined samples, an overall contamination rate of 7.4% can be regarded as more than satisfactory. In Cluster R (raw plant material and peat), the low sample contamination rate of only 2.6% is an excellent result. Even the highest contamination rate, seen in samples from Cluster E (animals’ faeces) at 12.6%, is considered acceptable (Figure 1). In none of the three clusters did the level of contamination exceed mycobacterial growth (Figure 2).

The degree of media contamination and mycobacterial recovery is comparable to the two-phase method used to treat avian guano with NaOH and cetypyrimidium chloride. Cultivation took place on soils enriched with antibiotics. Cultivation took place on antibiotic-enriched media with contamination reaching 19% and mycobacteria isolation reaching 27.8% [61]. In our study, we found a similar culture positivity, which reached 29.4% (Figure 2). In contrast, using our modified decontamination method, we recorded only 12.6% contamination of the samples (Figure 1).

In contrast to a study by Neumann et al. [62] comparing decontamination procedures for the treatment of water samples before mycobacterial isolation, a higher proportion of contamination was observed on media cultured at 30 °C. Moreover, in that study, mycobacterial growth was highest at 37 °C. On the other hand, in our study, both slow-growing and fast-growing mycobacteria were isolated especially in the mesophilic temperature range, i.e., at 30 °C (Table 1).

The differences in the spectrum of captured species of mycobacteria within individual clusters, particularly the higher level of detection in cluster R (peat and plant material), can be explained by known ecological characteristics [42]. Mycobacterial species are more commonly found in these matrices (Table 1). From an epidemiological point of view, these matrices, in which these mycobacterial species multiply, represent a natural reservoir [42,63]. Clusters S and E are likely vectors for the spread of mycobacterial species, or possibly only transient habitats in which they do not multiply (Table 1). However, it is possible that the multiplication of these mycobacterial species occurs to a limited extent under certain conditions, such as temperatures between 18 and 20 °C, pH below 7.0, and the presence of some organic substances as a source of nutrients [42,64,65].

Microscopic examination was only useful to evaluable samples containing a high concentration of mycobacterial cells. The presence of large amounts of solid residues and acid-fast-stained (i.e., red) artefacts make microscopic examination difficult to assess specifically in terms of objectivity (Figure 8). Microscopic examination can be replaced by the determination of the presence of mycobacterial DNA by qPCR. For practical reasons, a commercial product with a standardised protocol and a simple design was used in our study for the examination. The design for this *Mycobacterium* assay is based on the publication of Radomski et al. [20]. The results of the qPCR assay indicated a degree of discrepancy in the detection of the 16S RNA target site using a specific primer 110F/I571R with true capture and growth rate of live mycobacteria.

The primary evaluation of sensitivity and specificity of the PMA-qPCR method compared to the culture determination (considered the gold standard) showed values of 92.2% and 49.9%, respectively (Table 4). The presence of residues of genetic material from dead mycobacterial cells is the most likely cause of false positives seen in the PMA-qPCR method [42]. It is also not possible to exclude the interaction of primers with DNA segments of several representatives of the genera *Corynebacterium*, *Bacillus* and *Flavobacterium* very abundant in many matrices. This non-specific detection in the listed genera has also been reported by the authors of the first report of the qPCR method for detection of mycobacteria [20]. Alternatively, the PMA-qPCR method may be detecting the presence of unculturable mycobacterial species. Cut-off value derived from the model was determined to be 1.75 × 10^3^ mycobacterial cells, as measured by the qPCR method. After application of this cut-off value on the raw data, the diagnostic sensitivity decreased to 78.8%, but there was a significant increase in specificity to 77.3% (Table 5).

The work was performed continuously and retrospectively on natural matrices. We therefore accept that the unambiguous determination of the yield could be skewed by the difficulty of determining the true measurement of the presence of viable mycobacteria in the examined samples (missed blank samples). Yield can be affected of course by the decontamination method due to sensitivity of some mycobacterial species to a decontamination agent [66]. Cultivation may be (and will be in further work) refined by using special cultivation media (e.g., RGM) [67]. For detection of mycobacteria by standard qPCR in these samples, the specificity was not high, perhaps unsurprisingly, as this method was primarily designed for less microbially loaded samples. However, a more fundamental shift in detection accuracy occurred with the elimination of the DNA from dead cells after treatment of the sample with PMA before the qPCR. The average decrease in detection in the order of copies was −2 LOG, with sensitivity at 100% and specificity at 80% (shown in Table 6 and Figure 8), which is comparable with the values determined by the authors of the concept of this method for quantification of mycobacteria by qPCR [34].

## 5. Conclusions

For epidemiology studies, current methods for establishing the presence of mycobacteria in environmental matrices with high organic and microbial content are laborious, inaccurate, suffer from high levels of non-mycobacterial contamination, and are not amenable to high-throughput processing. We propose an improved method involving a modified decontamination process and a novel qPCR method incorporating a pre-treatment with propidium monoazide (PMA) to accurately detect the presence of viable mycobacteria in environmental samples. This method also allows for the quantification of the level of mycobacteria present and subsequent identification of individual mycobacterial species. Pre-processing environmental samples with decontamination solution containing 4.0% NaOH and of 0.5% tetradecyltrimethylammonium bromide (TDAB) was demonstrated to be effective and required the lowest amount of laboratory intervention compared to other commonly used methods. The addition of TDAB proved to be effective in increasing mycobacteria yield and eliminating gross contamination to 2–13% in three different kinds of heavily contaminated environmental matrices. A quantitative polymerase chain reaction (qPCR) method was used to detect the presence of mycobacteria in the decontaminated environmental samples. This optimised process was used for a clinical epidemiological human and veterinary study where environmental matrix samples heavily contaminated with organic matter were collected between January 2019 and March 2021. The improved method was successful in terms of resulting in a low rate of contamination, and processing was completed in an acceptable timeframe. The PMA qPCR was demonstrated to be as a suitable method for the detection of viable and uncultivable mycobacteria present in various matrices. Mycobacteria were cultured from 28.5% (Cluster S), 60.2% (Cluster R), and 29.4% (Cluster E). A total of 38 species of slowly and rapidly growing mycobacteria were isolated. *M. avium* ssp. *hominissuis*, *M. fortuitum* and *M. malmoense* were the most commonly isolated species. A simple decontamination procedure can facilitate and speed up laboratory work, and it can also be used in socio-economically stressed areas of the world for monitoring the occurrence of mycobacteria in the external environment. The method is applicable (at the same time it was introduced at the author’s workplace) in clinical practice for processing samples of clinical material with a high microbial load (stool, urine, and purulent samples). The acceptable sensitivity of the qPCR method has been confirmed and can be used in epidemiologically important areas (mining industry, clinical facilities) where rapid and continuous monitoring of mycobacterial exposure is needed. With the help of PMA-qPCR, it would be possible to realistically monitor the effectiveness of remediation and disinfection procedures in a short time.

## Data Availability

Availability of data and material correspondence and requests for mycobacterial isolates should be addressed to the corresponding author.

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
