# Peer review of "Recovery of Mycobacteria from Heavily Contaminated Environmental Matrices"

_microorganisms, 2021, doi:10.3390/microorganisms9102178_

Round 1
Reviewer 1 Report
Abstract:
Line 16: correct the grammar in this sentence.
Line 24: Define PMA.
Introduction
Line 34 “……………pathogenic and non-tuberculous mycobacteria (NTM)”. Readers might wrongly infer from this sentence that NTM are non-pathogenic. Several species are pathogenic. Please re-phrase.
Line 38: the authors state that: “……….decontamination with a sodium hydroxide solution requires a long incubation step which was not suitable for these types of samples where the processing for this study was required to be completed within 24 hours of collection”. Decontamination with sodium hydroxide typically takes < 30 minutes. The authors cite reference 2 here, which documents a time of 20 minutes (in Table 1). The authors need to clarify why a decontamination time of < 30 minutes is not suitable for samples where processing needs to be completed within 24 h. A better justification for not simply using NaOH would seem to be provided in lines 331-334.
Line 45: this should be corrected to: Cetylpyridinium chloride
Lines 52 & 54: Use “QUAS” and not “QUAS’s”. QUAS is already plural.
Line 93: ‘Sample collection’ would be better or ‘collection of samples’.
Line 95: Add “Czech Republic” after “Moravia”.
Methods:
Line 155: ‘approach’
Lines 164 – 171: don’t use “CFU’s”. CFU is already plural. Check throughout the whole paper.
Line 166: “the number of media”
Line 171: do you mean “colonies” rather than “plaques”?
Line 179: correct the grammar.
Results
Line 196 states that: “The contaminating microflora was thoroughly examined by microscopy and growth morphology on blood agar”. Can the authors confirm that species ID was performed by PCR? (as suggested in the methods). I am not sure why microscopy and growth morphology on blood agar are mentioned if the main method for identification was PCR. It would not be possible to distinguish between some of these species (e.g. Bacillus and Paenibacillis) using microscopy and growth morphology alone. I would suggest using “spp.” rather than “sp.” as presumably more than one species was encountered from these genera. The occurrence of Clostridium is surprising given that the vast majority of Clostridia require anaerobic conditions for growth except for a few microaerophilic strains.
Line 216: I wonder if “yield of mycobacteria” would be a better title for the figure that “sample contamination rates” in order to distinguish from Fig. 1.
Line 232: italics for ‘terrae’
Line 253 “loud”? Do you mean “load”? (see also line 272)
Line 280: do these figures for sensitivity and specificity refer to culture or PCR? This should be clearer.
Discussion
Line 330: throughput
Lines 400 – 412. This section includes a discussion of false positive PCR results and suggests various causes including “detecting the presence of unculturable mycobacterial species”. It would seem to be relevant here to discuss the likelihood that some NTM were probably not captured by culture due to the toxic effects of decontamination reagents. This possibility should at least be discussed. It is well established that 4% NaOH can kill rapidly-growing isolates of NTM (e.g. see: https://www.mdpi.com/2076-2607/9/8/1597/pdf). Another point that is worthy of inclusion in the discussion is the possibility of culture without decontamination. Alexander et al cultured 223 environmental water samples on RGM medium (without decontamination) and only recovered 1 isolate of non-mycobacteria (https://journals.plos.org/plosone/article?id=10.1371/journal.pone.0247166). This medium is now commercially available as NTM Elite agar (https://www.biomerieux-diagnostics.com/ntm-elite-agar). It would be of great interest to know if this medium is capable of similar performance with the grossly contaminated samples that have been examined in this study. This could make a very interesting future project and could be identified as such.
Conclusions
Lines 442-448. This is a repetition of earlier sections and should be omitted to make this section more concise.
Author Response
We thank to the reviewer for comments improving the quality of manuscript.

Reviewer 2 Report
The manuscript entitled "Recovery of mycobacteria from heavily contaminated environmental matrices" (Manuscript ID; microorganisms-1399868) focuses on the application of an effective decontamination protocol for the cultivation of mycobacteria from substrates highly contaminated by other bacterial species. Although the contents of the article are certainly of interest to experts in the field, there are some issues to be considered to improve the manuscript with a view to future publication.
The introduction should be revised to provide an adequate background for those approaching the subject for the first time. In this sense, more information should be included about the decontamination treatments in use, with particular emphasis on the contaminated matrices under study. Furthermore, the first part of the introduction (lines 32-42) seems more appropriate for the materials and methods section.
Also the part of materials and methods must necessarily be implemented. Although the methods described are clear, there are some parts in the results of which the methodology used is not indicated (e.g. identification of contaminated tubes and non-mycobacterial strains identification).
Finally, the conclusions should indicate the practical implications that this study brings.
Other minor comments:
- Line 24: indicate the complete name of PMA solution;
- Delete "PART ONE" and PART TWO" from the titles;
- Line 117: indicate the complete name of TDAB;
- Lines 134 and 136: hsp65 must be written in italics;
- Line 189 and following: indicate in brackets the fraction of the sample corresponding to the percentages, not just the total number of items analyzed;
- Lines 232: M. terrae must be written in italics;
- Table 1: change "M. sp" with "Mycobacterium sp.";
- Table 4: change "TNTC" with ">100".
Author Response

(The authors gave the same response as above.)

Round 2
Reviewer 2 Report
The authors made the corrections requested above. However, there are still some aspects to be settled before editing the work for publication. Here is the list below.
- Line 5: correct the affiliation number.
- Lines 13-15: please, rephrase the sentence "represents a relatively simple and universal procedure for processing samples of very complicated, heavily microbially contaminated matrices; increasing mycobacteria yield and eliminating gross contamination".
- Line 15: "2–13%": is this percentage correct? What does it indicate?
- Line 37: please, delete "which has increased hugely in popularity".
- Line 62: please cite these papers after "The slow-growing M. marinum is the causal agent of chronic systemic infections in fish"
- Delghandi, M.R.; El-Matbouli, M.; Menanteau-Ledouble, S. Mycobacteriosis and Infections with Non-tuberculous Mycobacteria in Aquatic Organisms: A Review. Microorganisms 2020, 8, 1368. https://doi.org/10.3390/microorganisms8091368
- Mugetti, D.; Varello, K.; Pastorino, P.; Tomasoni, M.; Menconi, V.; Bozzetta, E.; Dondo, A.; Prearo, M. Investigation of Potential Reservoirs of Non-Tuberculous Mycobacteria in a European Sea Bass (Dicentrarchus labrax) Farm. Pathogens 2021, 10, 1014. https://doi.org/10.3390/pathogens10081014.
- Line 68: please, change "capture" with "detection".
- Line 70: please, change "is" with "lead to".
- Line 81: please, change "it is possible to use" with "the most used mediums are".
- Lines 84-86: please, add a reference.
- Line 91: please, rephrase the sentence "These are most often quaternary ammonium salts (QUAS).".
- Line 107: please, change "Molecular genetic" with "Biomolecular".
- Line 113: please, change "[21,22,16,23–26]" with "[16, 21-26]".
- Line 123: you have previously indicated quantitative pcr with the acronym q-PCR in line 108. Why you change the acronym in this line? If if you mean "viability dyes in conjunction with quantitative PCR", you have to clarify the concept better.
- Line 133: please, change "and PMA" with ", PMA".
- Line 183: "4 300 rpm/min": please, indicate this unit measures more correctly.
- Line 198: please, indicate "quantitative PCR" with the acronym.
- Line 214: please, delete "fulminant".
- Line 218: what is meant by End's agar? Specify, as there are no references to it.
- Lines 231-232: was an initial denaturation cycle performed?
- Line 470-472: "The presence of large amounts of solid residues and acid-fast stained (i.e. red) artefacts make microscopic examination difficult to assess specifically in terms of objectivity": if available, insert an image to support the claim.
Author Response
Dear Editor,
We would like to thank both reviewer for valuable comments and suggestions. In the second round (Reviewer 2) we have accepted his suggestions and added the Figure 8 (direct microscopy examination from one culture positive but heavily contaminated sediment sample). We have also added 5 references (in green colour). In the manuscript all references were re-numbered (in yellow are reference added during the first round of revisions).
We hope, that now the quality of manuscript is improved.
Kind regards
Ivo Pavlik
Response to Reviewer 2-Round 2 Comments
Point 1: Line 5: correct the affiliation number.
Response 1: corrected.
Point 2: Lines 13-15: please, rephrase the sentence "represents a relatively simple and universal procedure for processing samples of very complicated, heavily microbially contaminated matrices; increasing mycobacteria yield and eliminating gross contamination".
Response 2: corrected.
Point 3: Line 15: "2–13%": is this percentage correct? What does it indicate?
Response 3: corrected.
Point 4: Line 37: please, delete "which has increased hugely in popularity".
Response 4: corrected.
Point 5: Line 62: please cite these papers after "The slow-growing M. marinum is the causal agent of chronic systemic infections in fish"
- Delghandi, M.R.; El-Matbouli, M.; Menanteau-Ledouble, S. Mycobacteriosis and infections with non-tuberculous mycobacteria in aquatic organisms: A review. Microorganisms 2020, 8, 1368. https://doi.org/10.3390/microorganisms8091368
- Mugetti, D.; Varello, K.; Pastorino, P.; Tomasoni, M.; Menconi, V.; Bozzetta, E.; Dondo, A.; Prearo, M. Investigation of potential reservoirs of non-tuberculous mycobacteria in a European Sea Bass (Dicentrarchus labrax) Farm. Pathogens 2021, 10, 1014. https://doi.org/10.3390/pathogens10081014.
Response 5: corrected including added references (in green colour).
Point 6: Line 68: please, change "capture" with "detection".
Response 6: corrected.
Point 7: Line 70: please, change "is" with "lead to".
Response 7: corrected.
Point 8: Line 81: please, change "it is possible to use" with "the most used mediums are".
Response 8: corrected.
Point 9: Lines 84-86: please, add a reference.
Response 9: added following references (in green colour in the list of references)
Pena, J.A.; Ferraro, M.J.; Hoffman, C.G.; Branda, J.A. Growth detection failures by the nonradiometric Bactec MGIT 960 mycobacterial culture system. J. Clin. Microbiol. 2012, 50, 2092–2095.
Piersimoni, C.; Nista, D.; Bornigia, S.; Gherardi, G. Unreliable detection of Mycobacterium xenopi by the nonradiometric Bactec MGIT 960 culture system. J. Clin. Microbiol. 2009, 47, 804–806.
Stephenson, D.; Perry, A.; Appleby, M.R.; Lee, D.; Davison, J.; Johnston, A.; Jones, A.L.; Nelson, A.; Bourke, S.J.; Thomas, M.F.; De Soyza, A.; Lordan, J.L.; Lumb, J.; Robb, A.E.; Samuel, J.R.; Walton, K.E.; Perry, J.D. An evaluation of methods for the isolation of nontuberculous mycobacteria from patients with cystic fibrosis, bronchiectasis and patients assessed for lung transplantation. BMC Pulm. Med. 2019, 19(1):19.
Point 10: Line 91: please, rephrase the sentence "These are most often quaternary ammonium salts (QUAS).".
Response 10: corrected.
Point 11: Line 107: please, change "Molecular genetic" with "Biomolecular".
Response 11: changed.
Point 12: Line 113: please, change "[21,22,16,23–26]" with "[16, 21-26]".
Response 12: changed.
Point 13: Line 123: you have previously indicated quantitative pcr with the acronym q-PCR in line 108. Why you change the acronym in this line? If if you mean "viability dyes in conjunction with quantitative PCR", you have to clarify the concept better.
Response 13: corrected.
Point 14: Line 133: please, change "and PMA" with ", PMA".
Response 14: corrected.
Point 15: Line 183: "4 300 rpm/min": please, indicate this unit measures more correctly.
Response 15: corrected.
Point 16: Line 198: please, indicate "quantitative PCR" with the acronym.
Response 16: corrected.
Point 17: Line 214: please, delete "fulminant".
Response 17: corrected.
Point 18: Line 218: what is meant by End's agar? Specify, as there are no references to it.
Response 18: corrected.
Point 19: Lines 231-232: was an initial denaturation cycle performed?
Response 19: explained.
Point 20: Line 470-472: "The presence of large amounts of solid residues and acid-fast stained (i.e. red) artefacts make microscopic examination difficult to assess specifically in terms of objectivity": if available, insert an image to support the claim.
Response 20: Figure 8 was inserted (Line 481).